# Pharmacokinetic Characterization of Supinoxin and Its Physiologically Based Pharmacokinetic Modeling in Rats

**DOI:** 10.3390/pharmaceutics13030373

**Published:** 2021-03-11

**Authors:** Yoo-Kyung Song, Yun-Hwan Seol, Min Ju Kim, Jong-Woo Jeong, Hae-In Choi, Seung-Won Lee, Yoon-Jee Chae, Sunjoo Ahn, Young-Dae Gong, Kyeong-Ryoon Lee, Tae-Sung Koo

**Affiliations:** 1Laboratory Animal Resource Center, Korea Research Institute of Bioscience and Biotechnology, Ochang-eup 28116, Korea; yksong777@kribb.re.kr (Y.-K.S.); redbijou@kribb.re.kr (M.J.K.); egaria1105@naver.com (J.-W.J.); 2Graduate School of New Drug Discovery and Development, Chungnam National University, Daejeon 34134, Korea; stivgma@naver.com (Y.-H.S.); chi705@naver.com (H.-I.C.); tmddnjs0831@naver.com (S.-W.L.); 3College of Pharmacy, Woosuk University, Wanju-Gun 55338, Korea; yjchae1020@gmail.com; 4Therapeutics & Biotechnology Division, Korea Research Institute of Chemical Technology, Daejeon 34114, Korea; sahn@krict.re.kr; 5Innovative Drug Library Research Center, Department of Chemistry, College of Science, Dongguk University, Seoul 04620, Korea; ydgong@dongguk.edu

**Keywords:** phosphorylated p68, supinoxin, pharmacokinetics, physiologically based pharmacokinetic modeling

## Abstract

Supinoxin is a novel anticancer drug candidate targeting the Y593 phospho-p68 RNA helicase, by exhibiting antiproliferative activity and/or suppression of tumor growth. This study aimed to characterize the in vitro and in vivo pharmacokinetics of supinoxin and attempt physiologically based pharmacokinetic (PBPK) modeling in rats. Supinoxin has good permeability, comparable to that of metoprolol (high permeability compound) in Caco-2 cells, with negligible net absorptive or secretory transport observed. After an intravenous injection at a dose range of 0.5–5 mg/kg, the terminal half-life (i.e., 2.54–2.80 h), systemic clearance (i.e., 691–865 mL/h/kg), and steady state volume of distribution (i.e., 2040–3500 mL/kg) of supinoxin remained unchanged, suggesting dose-independent (i.e., dose-proportional) pharmacokinetics for the dose ranges studied. After oral administration, supinoxin showed modest absorption with an absolute oral bioavailability of 56.9–57.4%. The fecal recovery following intravenous and oral administration was 16.5% and 46.8%, respectively, whereas the urinary recoveries in both administration routes were negligible. Supinoxin was mainly eliminated via NADPH-dependent phase I metabolism (i.e., 58.5% of total clearance), while UDPGA-dependent phase II metabolism appeared negligible in the rat liver microsome. Supinoxin was most abundantly distributed in the adipose tissue, gut, and liver among the nine major tissues studied (i.e., the brain, liver, kidneys, heart, lungs, spleen, gut, muscles, and adipose tissue), and the tissue exposure profiles of supinoxin were well predicted with physiologically based pharmacokinetics.

## 1. Introduction

Nuclear p68 RNA helicase is considered a prototypical DEAD box family of RNA helicases [1]. P68 is aberrantly expressed and dysregulated in various solid tumors, including colorectal and lung cancer [2,3], and the phosphorylation of p68 is reported to promote cell proliferation by activating the transcription of multiple cancer-related genes, including cyclin D1 and c-myc [4]. Specifically, the tyrosine phosphorylation of p68 at Y593 promotes β-catenin nuclear translocation and mediates the growth factor-stimulated epithelial–mesenchymal transition [5]. Therefore, the phosphorylated-p68 RNA helicase is increasingly recognized to be involved in cancer progression and metastasis [6,7].

Supinoxin (RX-5902) is a first-in-class, orally bioavailable small molecule inhibitor targeting the Y593 phosphorylated-p68 RNA helicase, and is being developed by Rexahn Pharmaceuticals (Rockville, MD, USA) [8,9]. By interacting with p68-RNA helicase and inhibiting β-catenin dependent ATPase activity [8,10], supinoxin exhibits antiproliferative activity and suppression of tumor growth against a variety of in vitro and in vivo tumor models, including triple-negative breast cancer [10,11,12,13].

The preliminary results after two cycles of supinoxin therapy in a phase 1b/2a trial in patients with previously treated advanced triple-negative breast cancer indicated that in five of the 11 patients treated, the disease was stable, while one subject exhibited early antitumor activity (a tumor reduction of 18.2%) and the suggested dose appeared to be safe and well tolerated [14]. However, the pharmacokinetics of the compound have not yet been fully characterized. Jeong et al. briefly summarized the pharmacokinetic profile of the drug in rats [15] and a phase I trial reported the safety and tolerability of supinoxin [14,16], detailed information on the distribution and elimination kinetics of the drug is still lacking. Pharmacokinetic studies in experimental animals, especially rats, can fill this gap by providing useful reference data for drug development that are rarely assessed in early clinical trials. Such studies are thus crucial to understand further, the behavior of the drug in the human body.

Thus, in this study, we investigated the in vitro and in vivo (after intravenous and oral administration in rats) pharmacokinetic characteristics of supinoxin and evaluated its main elimination pathways and distribution to major tissues (i.e., the brain, liver, kidneys, heart, lungs, spleen, gut, muscles, and adipose tissue). Finally, physiologically based pharmacokinetic (PBPK) modeling was used to simulate and predict the tissue distribution of the drug.

## 2. Materials and Methods

### 2.1. Chemicals

Supinoxin (Appendix A) and DGG-200064, an internal standard, were obtained from Dongguk University (Seoul, Korea). Dimethyl sulfoxide (DMSO) was obtained from Sigma-Aldrich (St. Louis, MO, USA). Polyethylene glycol-400 (PEG 400), and Tween 80 were obtained from Samchun Chemicals (Pyeongtaek, Korea). HPLC-grade acetonitrile and methanol were purchased from TEDIA Inc. (Fairfield, OH, USA). All other reagents and chemicals used were of analytical grade. Rat plasma was anticoagulated with sodium heparin and was obtained from our laboratory.

### 2.2. Animals

Male Sprague Dawley (SD) rats, 6–7 weeks old, were purchased from Orient Bio Inc. (Seongnam-si, Korea) and were used in all in vivo studies. The animals were maintained at a temperature of 20–26 °C with a 12 h light/dark cycle and a relative humidity of 40–60%. The experimental protocols involving the animals used in this study were reviewed by the Animal Care and Use Committee of Chungnam National University (CNU-00576, Daejeon, Korea; approval date: 13 March 2015).

### 2.3. Transport of Supinoxin in Caco-2 Cells

The Caco-2 intestinal epithelial cell line (ATCC^®^ HTB-37™) purchased from American Type Culture Collection (ATCC, Manassas, VA, USA) was cultured and maintained in Dulbecco’s modified Eagle medium (DMEM) containing 10% fetal bovine serum, 100 U/mL penicillin, and 0.1 mg/mL streptomycin under humidified atmosphere of air containing 5% CO_2_ at 37 °C. Caco-2 cells were seeded at a density of 6 × 10^4^ cells/cm^2^ onto polyethylene membranes of transwell plates (Corning Costar, Cambridge, MA, USA). The medium was replaced on alternate days after seeding, and cell monolayers were used for the transport assay after approximately 21 days of culture. Trans-epithelial electrical resistance (TEER) values of the Caco-2 monolayers were measured with an EVOM epithelial tissue voltmeter (World Precision Instruments, Sarasota, FL, USA), and cells with TEER greater than 300 Ω were used for the experiments. Cell monolayers were first rinsed three times with transport buffer (Hank’s balanced salt solution supplemented with 10 mM HEPES) and preincubated for 30 min at 37 °C. To measure the bidirectional transport of the compound, a transport buffer containing 1 µM supinoxin was added to the respective donor compartments (the apical and basolateral sides for absorptive and secretory transport assays, respectively). Samples (200 µL) were collected at 30, 60, 90, and 120 min, and an identical volume of fresh buffer was replenished. The samples were then subjected to LC–MS/MS analysis and concentrations of the samples were determined. After the final sampling from both compartments, we measured the TEER of the monolayers to ensure no possible adverse effects induced by the experimental procedures and then estimated the apparent permeability coefficient of supinoxin from the compound’s time-dependent accumulation (dQ/dt) in the receiver compartments using the following equation: P_app_ = dQ/dt × 1/(A × C_0_)(1)
where dQ/dt is the rate of transport, A is the surface area of the filter, and C_0_ is the initial concentration in the donor chamber.

### 2.4. In Vivo Pharmacokinetic Study

#### 2.4.1. Intravenous or Oral Administration of Supinoxin to Rats

Supinoxin was dissolved in a vehicle (10% DMSO, 10% Tween 80, 40% PEG 400, and 40% saline). In single doses, the administration routes involved an intravenous bolus via the tail vein (*n* = 4–5) and an oral gavage dose (*n* = 4–5). The dosing volume was 2 mL per kg body weight, and the dosing range was 0.5, 1, and 5 mg/kg. Blood samples (200 µL) were collected from the jugular vein at 0.083 (intravenous only), 0.25, 0.5, 1, 3, 7, 10, and 24 h after dosing, using a heparinized syringe to ensure anticoagulation. During blood sampling, rats were placed in a restrainer (Nagai-CFS-1S; NMS, Tokyo, Japan). For the separation of plasma fractions, all blood samples were centrifuged at 17,600× *g* for 5 min. The samples were stored at −20 °C until analysis. When obtaining tissue concentrations to calculate the tissue-to-plasma partition ratio was necessary, samples of nine tissues (i.e., adipose, heart, muscle, kidney, brain, liver, spleen, gut, and lung) were collected at 0.33, 1, 3, 7, and 24 h after 5 mg/kg intravenous administration of supinoxin (*n* = 3).

#### 2.4.2. Determination of Urinary and Fecal Excretion

Supinoxin was administered to male SD rats (*n* = 3–4) at a dose of 5 mg/kg either intravenously (through the tail vein) or by oral gavage, following which rats were kept in metabolic cages and urine and feces samples were collected over the following time intervals: 0–2, 2–4, 4–6, 6–10, 10–24, 24–36, and 36–48 h for urine; 0–10, 10–24, 24–36, and 36–48 h for feces.

The metabolic cages were rinsed with distilled water, and residues were added to the urine samples at 48 h. To extract the supinoxin present in the feces, samples were homogenized with a blender mixer and shaken vigorously for 4 h with 50% methanol.

The amount of drug excreted in urine or feces (A_e_) and the dose fractions excreted unchanged in urine or feces (f_e_) were calculated as:A_e_ = C_obs, urine or feces_ × V_urine or feces_(2)
f_e_ = A_e_/Dose(3)

The urinary and fecal clearances of supinoxin were calculated as:CL_urine or feces_ = A_e, urine or feces_/AUC(4)

### 2.5. Pharmacokinetic Analysis

The peak concentration (C_max_) and the time to reach C_max_ (T_max_) were obtained directly from individual plasma concentration–time profiles. Linear trapezoidal rule was used to calculate the areas under the plasma concentration–time curve (AUC) and the first moment curve (AUMC), while the terminal half-life (T_1/2_) was calculated using 0.693/λ, where λ represents the slope of the log-transformed concentration–time profiles of the terminal phase. The systemic clearance (CL), mean residence time (MRT), and the volume of the distribution at the steady state (V_ss_) were calculated as follows: dose/AUC, AUMC/AUC, and MRT·CL, respectively. To calculate the absolute oral bioavailability (F) we divided the AUC obtained following oral administration by that obtained following intravenous administration of the respective dose.

The tissue-to-plasma partition coefficient (K_p_) for supinoxin was calculated by dividing the mean AUC_tissue_ by the mean AUC_plasma_ after administration. To obtain the aforementioned pharmacokinetic parameters, all plasma and tissue concentration–time profiles were analyzed using a noncompartmental analysis using the WinNonlin software (ver. 5.3; Pharsight, St. Louis, MO, USA).

### 2.6. Estimation of the Unbound Fraction of Supinoxin in the Plasma and Microsomes 

To determine the unbound fraction of supinoxin in the plasma, a protein binding study was carried out using the Rapid Equilibrium Dialysis (RED) method. Following the manufacturer’s protocol (Thermo Fisher Scientific, San Jose, CA, USA), supinoxin was added to a plasma aliquot to produce a final concentration of 1 µg/mL. A plasma aliquot containing supinoxin (200 µL) and buffer (350 µL) was added to the sample chamber and the buffer chamber, respectively. When necessary, supinoxin was added to the microsomal incubation mixture (0.5 mg of protein per mL in potassium phosphate buffer) to produce final concentrations of 1 µM. The plate was covered and agitated on an orbital shaker at approximately 250 rpm at 37 °C. Aliquots (25 µL) of the plasma and buffer were collected at 4, 8, 18, and 24 h, and the samples were subjected to LC–MS/MS analysis. The fraction of supinoxin bound to plasma protein was estimated as follows [17]:Percent bound (%) = [1 − (concentration in buffer/concentration in plasma)] × 100%(5)

### 2.7. Metabolic Stability of Supinoxin

In these experiments, the metabolic stability of supinoxin in rat liver microsomes (BD Biosciences, San Jose, CA, USA) was determined. The microsomal reaction mixture comprised the follows: liver microsomal protein (final concentration of 0.5 mg protein per mL reaction mixture) and an NADPH regenerating solution (1.3 mM NADP+, 3.3 mM glucose-6-phosphate, 0.4 U/mL glucose-6-phosphate dehydrogenase, and 3.3 mM MgCl_2_) in a 100 mM potassium phosphate buffer with a pH of 7.4. After 5 min preincubation of the reaction mixture in a water bath at 37 °C, the reaction was initiated by the addition of a supinoxin solution to obtain a final concentration of 1 µM. Aliquots (50 µL) of the mixture were sampled at 5, 15, 30, and 60 min after the initiation of the reaction. The reaction was terminated by adding a stop solution (200 µL of ice-chilled acetonitrile) to the sample immediately after collection. After mixing by vortexing and centrifugation of the mixtures at 15,000× *g* for 5 min, an aliquot (150 µL) of the supernatant was analyzed by LC–MS/MS assay. In addition, the metabolic stability of supinoxin was determined in the presence of UDP glucuronic acid under similar reaction conditions, except that uridine 5′-diphosphoglucuronic acid was added instead of the NADPH regenerating solution. Finally, metabolic stability of supinoxin was tested in concentration range of 0.1–10 μM to confirm that the concentration used in the study (1 μM) was below the K_m_ for the metabolic reaction. The amount of sample remaining was plotted against the reaction time to determine the metabolic rate constant of the reaction. Assuming that the substrate concentration was below the K_m_ for the metabolic reaction, the intrinsic clearance was calculated as follows:(6)$${\mathrm{CL}}_{\mathrm{int}}=\;\frac{A_{\mathrm{protein}}\times k_{e}}{f_{u,\mathrm{mic}}\cdot C_{\mathrm{protein}}}$$
where A_protein_ is the total amount of liver microsomal protein, k_e_ is the metabolic rate constant of the reaction, f_u,mic_ is the unbound fraction of the drug in the reaction mixture, and C_protein_ is the microsomal protein concentration (mg/mL) in the reaction mixture. In this calculation, an A_protein_ of 1790 mg of protein per kg for rats was used [18].

When necessary, the hepatic clearance of the compound was calculated using a well-stirred model as follows: (7)$${\mathrm{CL}}_{h}=\;\frac{Q\cdot {\mathrm{CL}}_{\mathrm{int}}\cdot f_{u,p}}{Q+{\mathrm{CL}}_{\mathrm{int}}\cdot f_{u,p}/R}$$
where Q is the hepatic blood flow, f_u,p_ is the unbound fraction of the drug in the plasma, and R is the blood-to-plasma partition ratio. In this calculation, a hepatic blood flow of 14.5 mL/min for 250 g rat [19] was used, and the blood-to-plasma partition ratio was assumed to be 1.

### 2.8. CYP Inhibition by Supinoxin

CYP inhibition by supinoxin was studied using Vivid^®^ CYP450 screening kits (Thermo Fisher Scientific, San Jose, CA, USA) following the manufacturer’s instructions. Briefly, Vivid^®^ substrates were dissolved in acetonitrile yielding a stock solution of 2 mM. Fluorescent assays were conducted at room temperature using Costar 96-well black plates (#3915; Corning Costar, Acton, MA, USA). Supinoxin stock solution was diluted in reaction buffer to yield a 2.5X concentration solution (final concentration of 10 µM); aliquots (40 μL) of the solution were added to the wells. Subsequently, aliquots (50 μL) of reaction mixture consisting of P450 Baculosomes^®^ Reagent and the NADPH regeneration system were also added to the wells and preincubated for 10 min, at room temperature, after which 10 μL of 1 mM NADP^+^ and Vivid^®^ substrate mixture were added.

After a 30-min incubation, fluorescence was detected in end-point mode using a Varioscan™ Flash multimode reader (Thermo Fisher Scientific, San Jose, CA, USA). Background fluorescence was obtained from wells not containing CYP isozymes and was withdrawn before further reaction. The excitation and emission wavelengths of 485 nm/530 nm for Vivid^®^ Green substrates, and 409 nm/460 nm for Vivid^®^ Blue substrates were used. The inhibition percentage of the compound was determined in comparison to the vehicle (i.e., DMSO)-treated control wells. For isozymes with > 50% inhibition at 10 μM, we determined the concentration-dependent inhibition of supinoxin to obtain the IC50 value at concentrations ranging from 0.59 to 10 μM.

### 2.9. PBPK Modeling and Simulation

Whole-body PBPK modeling and simulation of supinoxin pharmacokinetics were performed using Simcyp Animal (ver. 17; SimCYP Ltd. Sheffield, UK). The supinoxin model was built using the physicochemical properties and predicted in silico data (MedChem Designer; ver 5.5, Simulations Plus, Inc. Lancaster, CA, USA; MarvinSketch; ver 21.3, ChemAxon Ltd. Budapest, Hungary) of supinoxin as well as in vitro and in vivo data obtained from this study (Table 1). In modeling the absorption process, the Advanced Dissolution, Absorption, and Metabolism (ADAM) model was employed with an effective permeability (P_eff,man_) of 1.317 × 10^−4^ cm/s, which was predicted by the Caco-2 permeability obtained from this study. In modeling the distribution process, a full PBPK model was selected, and the tissue-to-plasma partitioning coefficients (K_p_) obtained via in vivo experiments in this study (i.e., from adipose tissue, heart, muscle, kidney, brain, liver, spleen, gut, and lung samples) were used. The partitioning coefficients for tissues not addressed in this study (i.e., bones and skin) were predicted using the Rodgers and Rowland method [20]. In elimination modeling, we used the in vivo clearance obtained from the intravenous administration study.

### 2.10. Analytical Procedure for the Determination of Supinoxin Concentrations

To determine supinoxin concentrations in plasma, feces, and urine, we used a previously reported LC–MS/MS assay using DGG-200064 as an internal standard (IS) [15]. Briefly, an aliquot (50 μL) of IS solution (concentration 100 ng/mL) was added to 50 μL of plasma, feces, or urine sample, following which 400 μL of acetonitrile containing 0.1% formic acid was added. After vigorous vortexing for 10 min, the extract was centrifuged at 17,600 rpm for 10 min at 4 °C, and the supernatant was placed in an autosampler vial. An aliquot of 5 µL was then injected into the analytical column for chromatographic separation; the analytical system consisted of a 1200 series HPLC system (Applied Biosystems, Foster City, CA, USA) equipped with a turbo-electrospray interface in positive ionization mode.

Compounds were separated using a Zorbax XDB-C18 column (50 × 2.1 mm i.d., 3.5 µm; Agilent Technologies), and a mixture of 0.1% formic acid in distilled water and 0.1% formic acid in acetonitrile [50:50 (*v*/*v*)] at a flow rate of 0.3 mL/min was used as the mobile phase. Quantification was carried out using multiple reaction monitoring (MRM) at m/z 442.30 → 223.20 for supinoxin and m/z 430.08 → 223.20 for the IS. The optimized instrument conditions for the compounds were as follows: source temperature, 500 °C; curtain gas, 20 psi; nebulizing (GS1), 40 psi; heating (GS2), 40 psi; collision energy (CE), 23 V; entrance potential (EP), 10 V; collision energies (CEs), 23 V; collision cell exit potential (CXP), 14 V. In this study, the lower limit of quantitation (LLOQ) was defined as 0.5 ng/mL and intra-/interday precision (i.e., less than 13.7%) and accuracy (i.e., less than 11.6%) were found to be within the acceptance criteria for US Food and Drug Administration’s assay validation guidelines [21], indicating that the assay was valid within the concentration range studied.

### 2.11. Statistical Analysis

All data are presented as means ± SD. To compare the mean values between or among groups, the unpaired *t*-test or one-way analysis of variance (ANOVA) with Tukey’s post-hoc test were used. *p* values < 0.05 were considered statistically significant.

## 3. Results

### 3.1. Transport of Supinoxin in Caco-2 Cells

In this study, the permeability of supinoxin was estimated in Caco-2 cell monolayers. The apical to basolateral and basolateral to apical apparent permeability coefficients of supinoxin were estimated to be 18.5 ± 2.4 and 22.6 ± 1.8 × 10^−6^ cm/s, respectively. Comparably, the apparent permeability coefficient of metoprolol (i.e., high permeability control) was 15.5 ± 0.8 and 15.7 ± 0.8 × 10^−6^ cm/s for the apical to basolateral and basolateral to apical directions, respectively. The contribution of absorptive or secretory active transport was considered marginal, as the efflux ratio of supinoxin was calculated to be 1.24.

### 3.2. In Vivo Pharmacokinetic Study

The plasma concentration–time profiles and pharmacokinetic parameters of supinoxin were examined after the intravenous (Figure 1, Table 1, and Appendix A,) and oral (Figure 2, Table 2, and Appendix A) administration of doses ranging from 0.5 to 5 mg/kg in rats. Overall, pharmacokinetic parameters including the absolute oral bioavailability, V_ss_, CL, and the dose normalized C_max_ and AUC were not statistically different between doses. The cumulative amounts of supinoxin in urine and feces investigated after the intravenous and oral administration of a dose of 5 mg/kg in rats are shown in Table 3. The total urinary and fecal recovery of unchanged supinoxin after intravenous administration were 0.03% and 16.5%, respectively. After oral administration, the urinary and fecal recoveries were 0.01% and 46.8%, respectively. The lowest and highest mean AUC_24h_ for supinoxin in the tissues (i.e., brain, liver, kidneys, testes, heart, lungs, spleen, gut, muscles, and adipose tissue) were detected in the spleen (3.31 h·µg/mL) and adipose tissues (111 h·µg/mL), respectively, after the oral administration of a dose of 5 mg/kg in rats (Table 4). The tissue to plasma AUC ratio (K_p_) ranged from 0.65 (spleen) to 21.7 (adipose) in rats.

**Table 4 pharmaceutics-13-00373-t004:** Distribution of supinoxin after oral administration (5 mg/kg) and tissue volume of rat.

	AUC_0–24h_ ^a^ (µg/mL·h)	K_p_	Volume ^b^ (mL/kg)	K_p_ × Volume (mL/kg)
Plasma	5.108	1.00	31.2	31.2
Brain	4.349	0.85	6.80	5.80
Liver	24.31	4.76	41.0	196
Kidneys	11.13	2.18	9.20	20.0
Testes	5.370	1.05	10.0	10.5
Heart	3.551	0.70	3.20	2.20
Lungs	5.523	1.08	4.00	4.30
Spleen	3.314	0.65	2.40	1.60
Gut	25.31	4.96	40.0	198
Muscles	6.134	1.20	488	586
Adipose tissue	110.6	21.7	40.0	866
V_d_ (mL/kg)				1921.6

^a^ Mean value. ^b^ Data obtained from literature. Adapted from [22,23], Davies, B. et al., 1993 and Peters, S.A. et al., 2012.

### 3.3. Determination of the Plasma Protein Binding of Supinoxin

In this study, the extent of supinoxin binding to rat plasma protein was examined using the equilibrium dialysis method. During the 24 h incubation, the unbound fraction of supinoxin increased steadily up to 18 h, after which the fraction remained constant (Appendix A). After 18 h of incubation, the unbound fraction of supinoxin in rat plasma was 3.63 ± 0.25% at a 1-µg/mL concentration, suggesting that it was extensively bound to rat plasma protein. Similarly, the unbound fraction of supinoxin was also determined in a microsomal incubation mixture (0.5 mg of protein per mL). The unbound fraction was 62.3 ± 1.6% at a 1-µM concentration after 18 h of incubation (Appendix A).

### 3.4. Metabolic Stability 

For the evaluation of supinoxin metabolism, NADPH-dependent metabolic stability studies were carried out in rat liver microsomes (Figure 3A). The remaining amount at the end of the 60-min incubation was 12.7 ± 0.1% of the initial, the NADPH-dependent metabolic half-life of supinoxin was estimated to be 17.1 min, and the intrinsic clearance was 14.0 L/h/kg. Using a well-stirred model, hepatic clearance was estimated to be 443 mL/h/kg, accounting for 58.5% of the total in vivo clearance (i.e., the average clearance value for a 0.5–5 mg/kg intravenous administration was used for comparison). Supinoxin appeared to be stable for UDPGA-dependent metabolism, as the remaining amount was 92.2 ± 0.1% in the UDPGA-dependent metabolic stability study (Figure 3B). In addition, the addition of UDPGA in the NADPH-containing incubation mixture did not lead to significant metabolic profile differences from the mixture containing only NADPH (Figure 3A,C), indicating the negligible contribution of UDPGA-dependent metabolism in the microsomal clearance of supinoxin. Furthermore, the half-life of supinoxin remained constant in concentration ranges of 0.1–1 μM (Figure 3D), supporting the assumption that the concentration used in this study (1 μM) was below the K_m_ for the metabolic reaction.

### 3.5. CYP Inhibition Potency of Supinoxin

In the inhibition study, human recombinant CYP enzymes were used for the prediction of the drug–drug interaction potential in humans. Respective fluorescent substrates for CYP isozymes were used. Supinoxin caused little to no inhibition of CYP1A2, 2C9, 2C19, and 2D6 with 5.90%, 16.2%, 35.7%, and <0.01% inhibition, respectively, compared to the vehicle control, at a final concentration of 10 µM. As the inhibition was less than 50% at the highest concentration (i.e., 10 µM), the IC_50_ values for these isozymes could not be determined in the concentration range used, thus were inferred to be greater than 10 µM. In contrast, the inhibition of CYP3A4 was >50% at 10 µM of supinoxin. Therefore, a concentration-dependent inhibition study of supinoxin on CYP3A4 was conducted, resulting in an IC50 value of 4.66 µM (Appendix A).

### 3.6. Whole-Body PBPK

Whole-body PBPK modeling and simulation of supinoxin pharmacokinetics were performed using the pharmacokinetic parameters obtained from this study and the predicted in silico data (Table 5). Model parameters were refined, resulting in reasonable predictions for the concentration-profiles of supinoxin in the plasma and tissues (i.e., the brain, liver, kidneys, heart, lungs, spleen, gut, muscles, and adipose tissue) (Figure 4). The predicted-to-observed ratios of all the pharmacokinetic parameters, including T_max_, C_max_, AUC, and CL met the acceptance criterion of ranging between 0.5 and 2.

## 4. Discussion

Supinoxin was developed as an orally available inhibitor of phosphorylated P68. Although Jeong et al. recently provided a brief summary of the pharmacokinetic profile of the drug and a phase I trial reported the safety and tolerability of supinoxin [16], detailed analysis on the distribution and elimination kinetics of the drug is currently unavailable. In this study, the in vitro and in vivo pharmacokinetics of supinoxin were characterized in rats, using physiologically based pharmacokinetic analysis to predict the tissue distribution of the drug.

After the intravenous administration of supinoxin (0.5–5 mg/kg), the CL and V_ss_ values were consistent throughout the different doses, suggesting dose-independent (i.e., dose-proportional) pharmacokinetics of supinoxin. The compound was also shown to exhibit modest oral absorption with an oral bioavailability of 56.9–57.4%. Likewise, no statistically significant difference in oral bioavailability was observed between the different doses (0.5–5 mg/kg).

Caco-2 cell transport studies demonstrated that supinoxin is highly permeable with no apparent contribution from efflux transporters. The apparent permeability coefficient estimated in Caco-2 cells well correlated with human intestinal absorption [24], thereby suggesting an efficient absorption of supinoxin in the human intestine.

The distribution of supinoxin following oral administration was most abundant in the adipose tissue, followed by the gut and liver. Using the estimated K_p_ value and real tissue volumes from literature [22,23], the steady state volume of distribution was calculated as 1921.6 mL, which is reasonably close to the volume of distribution estimated in the noncompartmental analysis of the systemic administration study (2110 ± 803, 2040 ± 316, and 3500 ± 1450 mL/kg following the intravenous administration of 0.5, 1, and 5 mg/kg, respectively), suggesting that supinoxin is primarily distributed to the tissues examined in this study. In addition, we used PBPK modeling to understand the distribution characteristics of supinoxin in rats after oral dosing. For the PBPK model, we applied various pharmacokinetic parameters obtained in this study, in addition to the physicochemical properties of supinoxin. The simulated concentration profiles of supinoxin in the plasma and tissues were reasonably well matched with the observed values (Figure 4). The PBPK models used in this study appeared useful in predicting the plasma and tissue concentrations of supinoxin in rats. A similar PBPK strategy may also be used to predict pharmacokinetics in humans, such as predicting the optimal dose for clinical trials, the application when the modulation of the dose regimen or target population is necessary, or the concentration of the drug accumulated in a specific tissue.

To characterize the elimination kinetics of supinoxin, the urinary and fecal clearance of the compound were determined. As the systemic pharmacokinetics of supinoxin suggested no evidence of a saturable (i.e., nonlinear) process involved, the urinary and fecal excretions of the compound were determined by a representative single dose (5 mg/kg). The total urinary recovery of unchanged supinoxin following both intravenous and oral administration was less than 0.1% of the given dose, indicating that urinary excretion was negligible. The fecal recovery of supinoxin following intravenous and oral administration was 16.5% and 46.8%, respectively, the discrepancy possibly reflecting the amount of drug left unabsorbed through the GI tract. The fecal clearance of supinoxin was estimated to be 135 mL/h/kg. In contrast, the hepatic clearance of supinoxin, estimated by rat liver microsomal incubation, was 443 mL/h/kg, accounting for 58.5% of the total in vivo clearance. Thus, supinoxin may be primarily eliminated through hepatic metabolism with a minor contribution from fecal elimination in rats.

Finally, when an inhibition study of supinoxin was carried out in human recombinant enzymes to predict drug–drug interactions, supinoxin was unlikely to be a significant inhibitor of human CYP1A2, 2C9, 2C19, and 2D6 (i.e., IC_50_ > 10 µM). In contrast, significant in vitro inhibition was observed for CYP3A4, with an IC50 value of 4.66 µM. However, according to the preliminary data from phase I dose-escalating clinical trials of supinoxin, C_max_ was observed to be 99.1–707 μg/L (0.22–1.60 μM) in the dose range of 25–775 mg [16]. The US Food and Drug Administration recommends that for inhibitor drugs with an [I_max,u_] value (the maximal unbound concentration in plasma) divided by the K_i_ value greater than 0.02 or an [I_gut_] value (the maximal concentration in gastrointestinal tract; dose divided by 250 mL) divided by the K_i_ value greater than 10, should be further evaluated for possibility of clinical drug–drug interactions [25]. Based on data from the dose-escalating studies and assuming the unbound fraction of the drug in human plasma is comparable to that in rats (i.e., unbound fraction 0.0363), the [I_max,u_] or [I_gut_] divided by K_i_ is less than 0.02 or 10 (0.012 and 2.51, respectively). Therefore, the possibility of significant drug–drug interactions by major CYP enzymes, including CYP1A2, 2C9, 2C19, and 2D6 are considered unlikely, while CYP3A4-mediated interactions should be closely monitored with regard to clinical plasma concentrations at high doses in humans.

## 5. Conclusions

The pharmacokinetics of supinoxin in rats were characterized after the intravenous and oral administration of doses in the 0.5–5 mg/kg range. Supinoxin exhibited dose-independent pharmacokinetics in the dose ranges tested, appeared to be highly permeable, and showed modest bioavailability. Additionally, supinoxin appeared to be primarily eliminated by hepatic metabolism, with minor amounts eliminated by the fecal route. Supinoxin was distributed primarily in the adipose tissue, gut, and liver, and the distribution kinetics were well correlated with the PBPK simulation results developed in this study. Based on inhibition studies, the possibility of CYP1A2, 2C9, 2C19, and 2D6-mediated drug–drug interactions was estimated to be low for humans, while CYP3A4-mediated interactions should be closely monitored with regard to clinical plasma concentrations. The results of this study provide insights into the development and understanding of the compound.

## Figures and Tables

**Figure 1 pharmaceutics-13-00373-f001:**
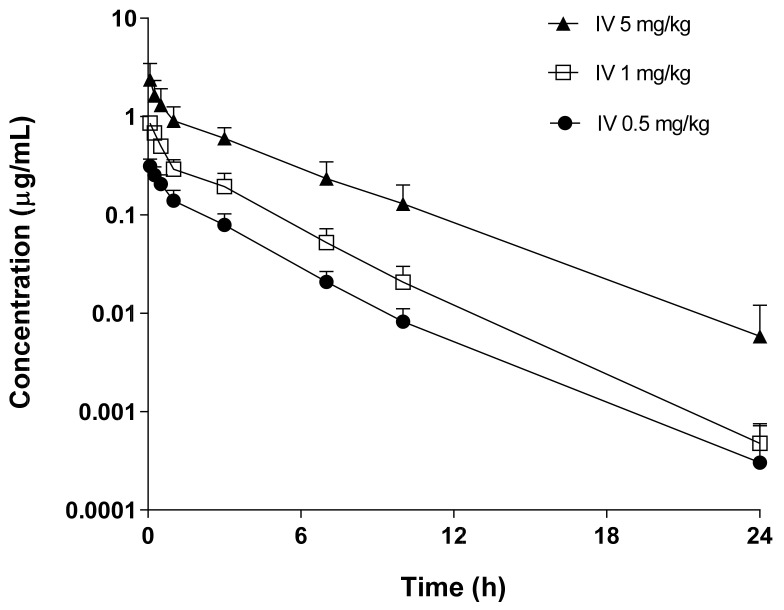
Pharmacokinetic profile of supinoxin following the intravenous administration of 0.5, 1, and 5 mg/kg to rats. Each point represents the mean ± standard deviation (*n* = 4–5).

**Figure 2 pharmaceutics-13-00373-f002:**
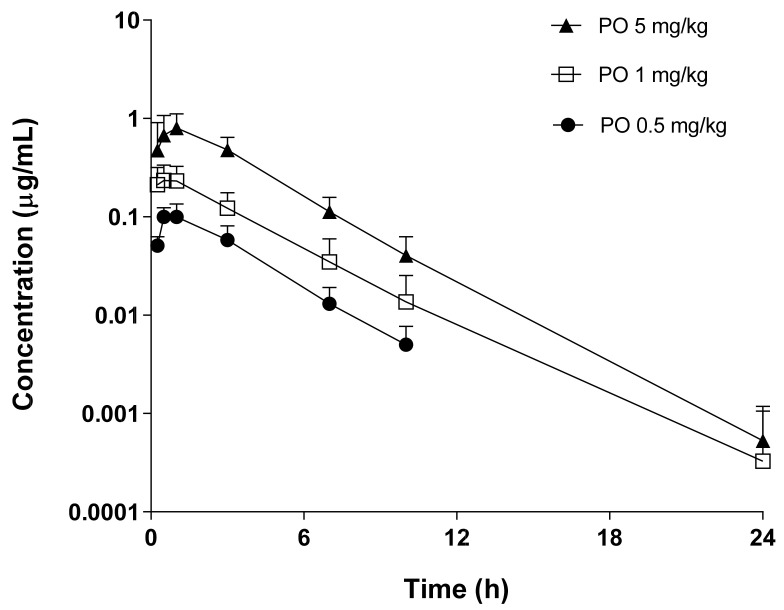
Pharmacokinetic profile of supinoxin following the oral administration of 0.5, 1, and 5 mg/kg in rats. Each point represents the mean ± standard deviation (*n* = 4–5).

**Figure 3 pharmaceutics-13-00373-f003:**
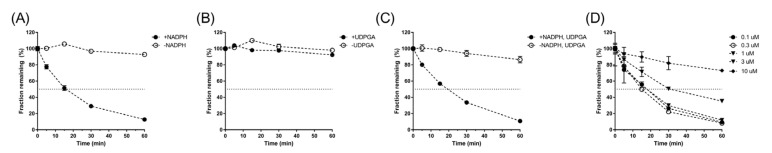
Metabolic stability of supinoxin in rat liver microsomes. Rat liver microsomes were incubated with 1 μM supinoxin for 60 min in the presence and absence of (**A**) NADPH, (**B**) UDP-glucuronic acid (UDPGA), or (**C**) both NADPH and UDPGA as a cofactor. In addition, (**D**) metabolic stability of supinoxin in concentration ranges of 0.1–10 μM was tested. Data are expressed as the mean ± standard deviation of triplicate runs.

**Figure 4 pharmaceutics-13-00373-f004:**
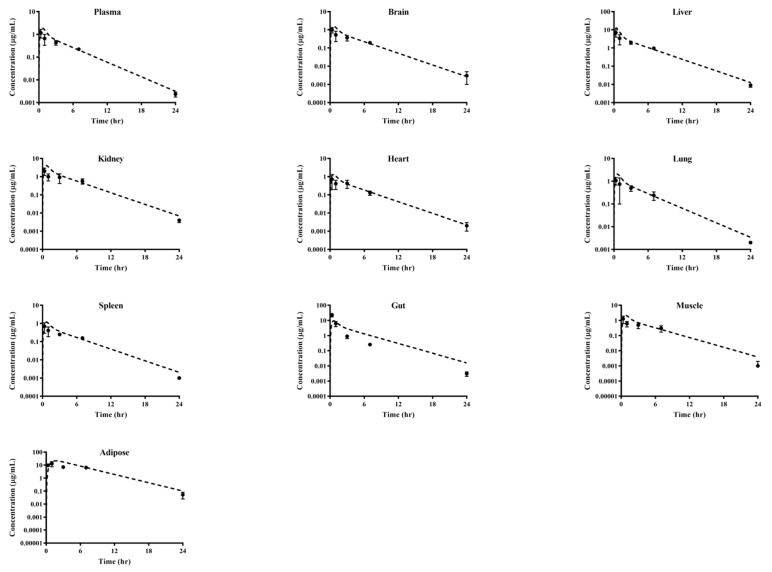
Observed and simulated plasma and tissue concentration–time profiles of supinoxin in different tissues after the oral administration of 5 mg/kg to rats. Closed squares and dotted lines represent the observed and simulated values, respectively (*n* = 3).

**Table 1 pharmaceutics-13-00373-t001:** Pharmacokinetic parameters of supinoxin after intravenous administration ^a^.

Parameter	Dose		
	0.5 mg/kg	1 mg/kg ^b^	5 mg/kg
AUC_last_ (µg·h/mL)	0.715 ± 0.180	1.75 ± 0.32	6.09 ± 1.76
AUC_inf_ (µg·h/mL)	0.727 ± 0.177	1.78 ± 0.30	6.13 ± 1.75
T_1/2_ (h)	2.54 ± 0.83	2.64 ± 0.39	2.80 ± 0.71
CL (mL/h/kg)	717 ± 152	691 ± 110	865 ± 235
MRT (h)	2.92 ± 0.65	2.99 ± 0.51	4.07 ± 1.37
V_ss_ (mL/kg)	2110 ± 803	2040 ± 316	3500 ± 1450

^a^ The data are represented as mean ± SD (*n* = 4–5). ^b^ This result has been previously reported. Adapted from [15], Jeong, J.-W. et al., 2017.

**Table 2 pharmaceutics-13-00373-t002:** Pharmacokinetic parameters of supinoxin after oral administration ^a^.

Parameter	Dose		
	0.5 mg/kg	1 mg/kg ^b^	5 mg/kg
AUC_last_ (h·µg/mL)	0.402 ± 0.130	0.984 ± 0.407	3.48 ± 1.45
AUC_inf_ (h·µg/mL)	0.418 ± 0.133	1.01 ± 0.42	3.51 ± 1.42
T_1/2_ (h)	2.00 ± 0.50	2.11 ± 0.91	2.02 ± 0.33
T_max_ (h)	0.800 ± 0.274	0.650 ± 0.337	0.750 ± 0.289
C_max_ (µg/mL)	0.108 ± 0.036	0.250 ± 0.093	0.823 ± 0.340
MRT (h)	2.99 ± 0.71	3.15 ± 1.26	3.17 ± 0.36
F (%)	57.4 ± 18.3	56.9 ± 23.7	57.2 ± 23.1

^a^ The data are represented as mean ± SD (*n* = 4–5). ^b^ This result has been previously reported. Adapted from [15], Jeong, J.-W. et al., 2017.

**Table 3 pharmaceutics-13-00373-t003:** Amount of drug excreted and unchanged fraction of excreted dose after intravenous or oral administration in rats (5 mg/kg).

	IV (*n* = 4)		PO (*n* = 4)	
	Amount (μg)	Fraction Excreted (%)	Amount (μg)	Fraction Excreted (%)
Urinary excretion	0.27 ± 0.06	0.03 ± 0.01	0.06 ± 0.04	0.01 ± 0.01
Fecal excretion	142 ± 21.6	16.5 ± 2.82	412 ± 209	46.8 ± 24.5

**Table 5 pharmaceutics-13-00373-t005:** Summary of input parameters for supinoxin in the PBPK model.

Parameters	Supinoxin	Source
**Physicochemical properties**		
Molecular weight (MW, g/mol)	441.465	
Log P_O:W_	2.7	MedChem Designer
Compound type	Monoprotic acid	
pK_a_	1.5	MarvinSketch
Blood to plasma partition ratio (B/P)	1	Assumed
Fraction unbound in plasma (f_u_)	0.0363	Determined
**Absorption**		
Absorption type	ADAM model	
Caco-2 permeability (10^−6^ cm/s)	18.47	Determined
**Distribution**		
Distribution model	Full PBPK model	
Vss (L/kg)	2.448	Predicted (Method2)
Tissue: Plasma partition coefficients	0.649–21.7	Determined
**Elimination**		
In vivo clearance (CL_iv_, mL/min)	3.16	Determined

## Data Availability

Not applicable.

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
