# Peer review of "Pharmacokinetic Characterization of Supinoxin and Its Physiologically Based Pharmacokinetic Modeling in Rats"

_pharmaceutics, 2021, doi:10.3390/pharmaceutics13030373_

Round 1

Reviewer 1 Report

The manuscript is clear and easily followed. All methods are adequately describe. I have only minor comments. 
1) In part 2.4.1. you write that "When it was necessary to obtain tissue concentrations ...samples of nine tissues were collected" but I miss the information about number of rats. I think this information should be add.
2) I miss the information why there is no point at 24h in Figure 2. after PO 0.5 mg/kg.
3)  You studied the unchange fraction of excreted dose in fecal and urine. You also did metabolic stability and found that supinoxin is metabolized by liver microsomes. Did you also try to analyzed, how the metabolites are excreted? 

Author Response

Response to Reviewer 1 Comments

The manuscript is clear and easily followed. All methods are adequately describe. I have only minor comments. 

1) In part 2.4.1. you write that "When it was necessary to obtain tissue concentrations ...samples of nine tissues were collected" but I miss the information about number of rats. I think this information should be add.

Thank you for your comment. We have added the number of animals in the revised manuscript as follows:

“…When obtaining tissue concentrations to calculate the tissue-to-plasma partition ratio was necessary, samples of nine tissues (i.e., adipose, heart, muscle, kidney, brain, liver, spleen, gut, and lung) were collected at 0.33, 1, 3, 7, and 24 h after 5 mg/kg intravenous administration of supinoxin (n = 3).

2) I miss the information why there is no point at 24h in Figure 2. after PO 0.5 mg/kg.

Thank you for your comment. The points at 24 h were not marked since all objects (n = 5) were less than the LLOQ. The plasma concentration-time data have been added to the Supplementary Tables.

3)  You studied the unchange fraction of excreted dose in fecal and urine. You also did metabolic stability and found that supinoxin is metabolized by liver microsomes. Did you also try to analyzed, how the metabolites are excreted? 

Thank you for your comment and suggestion. We have only conducted the metabolic stability test, and not the metabolite identification test due to limited resources, therefore, we did not study the metabolites, thank you for understanding.

Reviewer 2 Report

The manuscript presents preclinical data and a PBPK approach regarding supinoxin pharmacokinetics. The work is interesting and well organized regarding the amount and type of the experiments employed to create the PBPK approach. I believe they may try to advance their PBPK approach further in virtual populations so I wish them best of luck if they do so. However, there are some issues that are confusing in this work mainly in results section and should be addressed prior to any consideration. 

  1. Please provide the chemical representation of supinoxin 
  2. "Assuming that the substrate concentration was below the Km for the metabolic reaction". Please explain how it was assumed or provide relative reference data.
  3. I would suggest that LC/MS/MS data to be placed in the supplementary file
  4. Please place the results in the same order as you place them in methods section because it makes the reading of the manuscript confusing. Please provide in results section, results data and in methods section only experimental procedures. The phrase "For CYP3A4, the concentration-dependent inhibition of supinoxin (final concentration 0.59–10 μM) was also determined to obtain the IC50 value" is probably derived from final analysis due to results not in methodological design of the experiments.
  5. Too much details in the 3.2 section. Since they repeat data from Tables maybe they should consider to simplify it.
  6. There is small confusion the dose of 1 mg/kg PK parameters. The reported ones here are the ones calculated in this study or in ref 15? If it is ref 15 it should be omitted from the results and discussed as comparison with the ones actually estimated here. If not please provide relative comparison of PK profiles between the two here or in supplementary.
  7. 3.2. "respectively, after the intravenous administration of a dose of 5 mg/kg in rats (Table 4)" Table 4 legend: Table 4. Distribution of supinoxin after oral administration (5 mg/kg) and tissue volume of rat. So what we see in Table 4?oral or iv results?
  8. "Whole-body PBPK modeling and simulation of supinoxin pharmacokinetics were performed using the pharmacokinetic parameters obtained from this study and the predicted in silico data (Table 5)." There is no mention for in silico data in methods. Please also reference the other software you used in methods section. As I understand Simcyp was not the only PBPK simulator applied in this study.
  9. Since the PBPK model was employed where are the results regarding the PBPK simulations? There is only figure 4 and Table 5 with in silico data. Did the model well-predicted the C-t profile? Perhaps comparison of the values simulated and in vivo in a table? Is still the whole-PBPK rat model of Simcyp with one rat or average rat population?

Author Response

Response to Reviewer 2 Comments

The manuscript presents preclinical data and a PBPK approach regarding supinoxin pharmacokinetics. The work is interesting and well organized regarding the amount and type of the experiments employed to create the PBPK approach. I believe they may try to advance their PBPK approach further in virtual populations so I wish them best of luck if they do so. However, there are some issues that are confusing in this work mainly in results section and should be addressed prior to any consideration. 

1. Please provide the chemical representation of supinoxin 

As suggested, the chemical structure of supinoxin, shown below, has been added as Supplementary Figure 1.

Supplementary Figure 1. Chemical structure of supinoxin             

2. "Assuming that the substrate concentration was below the Km for the metabolic reaction". Please explain how it was assumed or provide relative reference data.

Thank you for your thoughtful advice. In a separate test using rat liver microsomes, the half-lives of supinoxin at 0.1, 0.3, 1 μM did not show significant difference, supporting our assumption that 1 μM to be less than the Km value. In contrast, the half-lives were markedly longer at 3 and 10 μM. The manuscript and relevant figure was therefore modified as follows by adding those supportive evidence.

(In methods section)

Finally, metabolic stability of supinoxin was tested in concentration range of 0.1-10 μM to confirm that the concentration used in the study (1 μM) was below the Km for the metabolic reaction.

(In results section)

Furthermore, the half-life of supinoxin remained constant in concentration ranges of 0.1-1 μM (Fig. 3D), supporting the assumption that the concentration used in this study (1 μM) was below the Km for the metabolic reaction.

(Modified figure 3)

Figure 3. Metabolic stability of supinoxin in rat liver microsomes. Rat liver microsomes were incubated with 1 μM supinoxin for 60 min in the presence and absence of (A) NADPH, (B) UDP-glucuronic acid (UDPGA), or (C) both NADPH and UDPGA as a cofactor. In addition, metabolic stability of supinoxin in (D) concentration ranges of 0.1-10 μM was tested. Data are expressed as the mean ± standard deviation of triplicate runs.

3. I would suggest that LC/MS/MS data to be placed in the supplementary file

Thank you for your suggestion. Accordingly, the plasma concentration raw data was added to the Supplementary File.

4. Please place the results in the same order as you place them in methods section because it makes the reading of the manuscript confusing. Please provide in results section, results data and in methods section only experimental procedures. The phrase "For CYP3A4, the concentration-dependent inhibition of supinoxin (final concentration 0.59–10 μM) was also determined to obtain the IC50 value" is probably derived from final analysis due to results not in methodological design of the experiments.

Thank you for your suggestion. Accordingly, we have matched the order of each experiment in the methods and results sections. Also, in the revised methods section, irrelevant text was removed and we have made the following revision:

“…For isozymes with > 50% inhibition at 10 μM, we determined the concentration-dependent inhibition of supinoxin to obtain the IC50 value at concentrations ranging from 0.59 to 10 μM.….”

5. Too much details in the 3.2 section. Since they repeat data from Tables maybe they should consider to simplify it.

Thank you for your suggestion and we agree with your assessment. All individual parameters have been deleted and simplified as follows.

“The plasma concentration-time profiles and pharmacokinetic parameters of supinoxin were examined after the intravenous (Supplementary Figure 1, Fig. 1, and Table 1) and oral (Supplementary Figure 2, Fig. 2, and Table 2) administration of doses ranging from 0.5–5 mg/kg in rats. Overall, pharmacokinetic parameters including the absolute oral bioavailability, Vss, CL, and the dose normalized Cmax and AUC were not statistically different between doses.….”

6. There is small confusion the dose of 1 mg/kg PK parameters. The reported ones here are the ones calculated in this study or in ref 15? If it is ref 15 it should be omitted from the results and discussed as comparison with the ones actually estimated here. If not please provide relative comparison of PK profiles between the two here or in supplementary.

Thank you for your comment and question, and we agree with your assessment. Since the 1 mg/kg result has already been reported, we should have retested it, and we apologize for not doing so. However, please note that the 1 mg/kg data were not simply taken from other groups, rather, they were part of a series of experimental data reported by same authors at the same time. Previous studies have reported some of the experimental data to verify the analytical method used, and in this paper, the results of the pharmacokinetic properties observed were based on dose changes. As the reviewer suggested, if we delete the 1mg/kg data from the results and mentioned the discussion part as is, this paper structure will be very awkward. We have already indicated in the table that the date is from previous studies; we appreciate your understanding.

7. 3.2. "respectively, after the intravenous administration of a dose of 5 mg/kg in rats (Table 4)" Table 4 legend: Table 4. Distribution of supinoxin after oral administration (5 mg/kg) and tissue volume of rat. So what we see in Table 4? oral or iv results?

Thank you for your comment. Indeed, the tissue distribution study was based on the 5 mg/kg oral administration, therefore, we have revised the results as follows:

“…The lowest and highest mean AUC24h for supinoxin in the tissues (i.e., brain, liver, kidneys, testes, heart, lungs, spleen, gut, muscles, and adipose tissue) were detected in the spleen (3.31 h·µg/mL) and adipose tissues (111 h·µg/mL), respectively, after the oral administration of a dose of 5 mg/kg in rats (Table 4)….”

8. "Whole-body PBPK modeling and simulation of supinoxin pharmacokinetics were performed using the pharmacokinetic parameters obtained from this study and the predicted in silico data (Table 5)." There is no mention for in silico data in methods. Please also reference the other software you used in methods section. As I understand Simcyp was not the only PBPK simulator applied in this study.

Thank you for the reviewer's comment. We agree with the reviewer's opinion, and in accordance with the opinion, the source and use of in silico data is indicated in the method section as follows.

“…The supinoxin model was built using the physicochemical properties and predicted in silico data (MedChem Designer; ver 5.5, Simulations Plus, Inc. Lancaster, CA, USA and MarvinSketch; ver 21.3, ChemAxon Ltd. Budapest, Hungary) of supinoxin as well as in vitro and in vivo data obtained from this study (Table 1)….”

9. Since the PBPK model was employed where are the results regarding the PBPK simulations? There is only figure 4 and Table 5 with in silico data. Did the model well-predicted the C-t profile? Perhaps comparison of the values simulated and in vivo in a table? Is still the whole-PBPK rat model of Simcyp with one rat or average rat population?

We agree with the reviewer's opinions and appreciate the insightful questions. As the reviewer mentioned, we established the PBPK rat model with average rat data. Additionally, we added the comparison data using the scatter plot of all observed versus predicted values as added Supplementary Figure 3, which suggested the resultant prediction is good.

Supplementary Figure 3. Relationship between observed and predicted concentration.

Round 2

Reviewer 2 Report

The authors presented an updated version of their manuscript and fully addressed the original comments. The manuscript can be further proceed.